# The impact of injury of the chorda tympani nerve during primary stapes surgery or cochlear implantation on taste function, quality of life and food preferences: A study protocol for a double-blind prospective prognostic association study

Esther E. Blijleven[1,2]*, Inge Wegner[3], Robert J. Stokroos[1,2], Hans G. X. M. Thomeer[1,2]

1 Department of Otorhinolaryngology, Head and Neck Surgery, University Medical Center Utrecht, Utrecht, Utrecht, the Netherlands, 2 Brain Center, University Medical Center Utrecht, Utrecht, Utrecht, the Netherlands, 3 Department of Otorhinolaryngology, Head and Neck Surgery, University Medical Center Groningen, Groningen, Groningen, the Netherlands

* e.e.blijleven-2@umcutrecht.nl

## Abstract

### Background

The chorda tympani nerve (CTN) is a mixed nerve, which carries sensory and parasympathetic fibres. The sensory component supplies the taste sensation of the anterior two-thirds of the ipsilateral side of the tongue. During middle ear surgery the CTN is exposed and frequently stretched or sacrificed, because it lacks a bony covering as it passes through the middle ear. Injury may cause hypogeusia, ageusia or altered taste sensation of the ipsilateral side of the tongue. To date, there is no consensus regarding which type of CTN injury (sacrificing or stretching), during middle ear surgery, leads to the least burden for the patient.

### Methods

A double-blind prospective prognostic association study was designed in a single medical centre in the Netherlands to determine the effect of CTN injury on postoperative taste disturbance and quality of life. 154 patients, who will undergo primary stapes surgery or cochlear implantation will be included. The taste sensation, food preferences and quality of life of these patients will be evaluated preoperatively and at one week, six weeks and six months postoperatively using the Taste Strip Test, Electrogustometry, supplementary questionnaire on taste disturbance, Macronutrient and Taste Preference Ranking Task, Appetite, Hunger and Sensory Perception questionnaire and Questionnaire of Olfactory Disorders to assess the association of these outcomes with CTN injury. Evaluation of olfactory function will only take place preoperatively and at one week postoperatively using the Sniffin' Sticks. The patient and outcome assessor are blinded to the presence or absence of CTN injury.

**Data Availability Statement:** Deidentified research data will be made publicly available when the study is completed and published.

**Funding:** The authors received no specific funding for this work.

**Competing interests:** The authors have declared that no competing interests exist.

**Abbreviations: AHSP**, Appetite, Hunger and Sensory Perception questionnaire; **CTN**, Chorda Tympani Nerve; **eCRF**, electronic Case Report Form; **EGM**, Electrogustometry; **ENT**, Ear, Nose and Throat; **MTPRT**, Macronutrient and Taste Preference Ranking Task; **NTR**, Netherlands Trial Register; **QoD**, Questionnaire of Olfactory Disorders; **TST**, Taste Strip Test; **UMCU**, University Medical Center Utrecht.

## Discussion

This study is the first to validate and quantify the effect of chorda tympani nerve injury on taste function. The findings of this study may lead to evidence-based proof of the effect of chorda tympani injury on taste function with consequences for surgical strategies.

## Trial registration

Netherlands Trial Register NL9791. Registered on 10 October 2021.

## Introduction

The chorda tympani nerve (CTN), a branch of the facial nerve, is a mixed nerve, which carries sensory and parasympathetic fibres. The sensory component supplies the taste sensation of the anterior two-thirds of the ipsilateral side of the tongue. The parasympathetic component innervates the submandibular and sublingual salivary gland. The CTN passes the medial surface of the neck of the malleus and runs in between the malleus and the incus in the tympanic cavity. During middle ear surgery, the CTN is exposed and frequently stretched or sacrificed, because it lacks a bony covering as it passes through the middle ear. Injury may cause hypogeusia, ageusia or altered taste sensation of the ipsilateral side of the tongue. Patients may also suffer from a dry mouth [1].

To date, there is no consensus regarding which type of CTN injury, obtained during primary stapes surgery or cochlear implantation, leads to the least burden. Change in taste sensation can alter the enjoyment of food, which is a burden to the patient [2, 3]. Some articles concluded that the taste outcome is better if the CTN is sacrificed instead of being preserved but damaged by stretching [4–7]. Other articles report that sacrificing the CTN results in more taste disturbance compared to nerve preservation [8–11]. Lastly, Rice et al. concluded that sacrificing or preserving the CTN makes little difference for the patient. Sparing the CTN may hinder an adequate view of the middle ear structures during surgery. If in fact sacrificing the CTN leads to less or equal taste disturbance compared to stretching it, sparing the CTN would not be justified [12].

A published systematic review of the effect of CTN injury during non-inflammatory middle ear surgery concluded that more patients with a stretched CTN experienced hypogeusia, ageusia or altered taste sensation than patients with a sacrificed CTN. The included studies were characterized by several limitations in their study design. First, the follow-up duration varied greatly between the included studies with a range of 6 weeks to 99 months. A longer follow-up duration may result in fewer symptoms due to the longer recovery time or habituation of symptoms [1]. Secondly, age seems to be associated with recovery rate. Younger patients seem to have significantly higher recovery rates of CTN function compared to older patients [13]. Most of the included studies disregarded age. Thirdly, postoperative taste dysfunction may have been underreported. It is not clear whether surgeons routinely asked their patients about taste dysfunction in the included studies. Lastly, most studies did not include quality of life as an outcome measure and therefore it remains unclear whether taste dysfunction significantly impacts quality of life [1].

### Purpose of this study

A lack of high-quality studies on the subject precludes firm evidence-based recommendations and demonstrates the need for a high-quality study. In order to accommodate this need, the

effect of CTN injury on postoperative taste disturbance will be investigated using the Taste Strip Test (TST) in this double-blind prospective study. The secondary objective is the postoperative taste disturbance using Electrogustometry (EGM), supplementary questionnaire on taste disturbance and Appetite, Hunger and Sensory Perception questionnaire (AHSP), food preferences using the Macronutrient and Taste Preference Ranking Task (MTPRT) and quality of life using the Questionnaire of Olfactory Disorders (QoD). The olfactory function using the Sniffin' Sticks will also be evaluated.

## Materials and methods

### Study setting

The study concerns a double-blind prospective prognostic association study at the outpatient Ear, Nose and Throat (ENT) department of the University Medical Center Utrecht (UMCU). The expected completion date for data collection is January 2024. The study protocol is in accordance with the Standard Protocol Items: Recommendations for Interventional Trials (SPIRIT) [14].

### Primary objective

The primary objective is to investigate the effect of CTN injury on postoperative taste function, defined as the postoperative taste strip score, in patients undergoing primary stapes surgery or primary cochlear implantation.

### Sample selection

Patients who will undergo primary stapes surgery or primary cochlear implantation will be included. This study has no public involvement group. The patients have to meet the following inclusion criteria to be suitable for inclusion.

**Inclusion criteria.**

- Signed informed consent form;

- Age $\geq$ 18 years;

- Scheduled or on the waiting list for primary stapes surgery or cochlear implantation;

- Willingness and ability to participate in all scheduled procedures outlined in the study protocol;

- Good understanding of the Dutch language.

A potential subject who meets any of the following criteria will be excluded from participation in the study.

**Exclusion criteria.**

- Previous middle ear surgery of the ipsilateral ear (with the exception of the placement of ventilation tubes in childhood);

- Ear malformation;

- Disability that could interfere with taste evaluation and/or questionnaire fulfilment;

- History of radiotherapy of the head and neck or chemotherapy;

- History of ipsilateral facial nerve palsy;

- Cardiac pacemaker;

- Pregnancy;

- History of orofacial pain;

- History of dysesthesia in the orofacial region;

- Local evidence of a pathological condition of the oral mucosa.

## Sample size

Linear regression analysis for our primary outcome measure, which is taste function defined as the taste strip score identified postoperatively, will be performed. Including our potentially predictor covariates, a total of 12 variables will be included in the model. The variables will be injury of the chorda tympani nerve, age, gender, smoking, weight, gastroesophageal reflux, olfactory dysfunction, salivary hypofunction, medication, rhinosinusitis/allergic rhinitis, previous COVID-19 and diabetes mellitus [15–25]. These variables are 14 degrees of freedom in total. Ten patients per degree of freedom will be included. To anticipate withdrawal of 10% of the included patients, an extra 14 patients will be included which means a sample size of 154 patients will be needed.

## Patient recruitment

Patients will be recruited from the ENT department of the UMCU. At the ENT department, 80 otosclerosis patients undergo primary stapes surgery per year and 100 patients undergo cochlear implantation per year. Assuming a participation rate of 70%, 154 patients are expected to be included in 15 months' time. However, the inclusion of patients was delayed due to the COVID-19 pandemic. Eligible patients will be asked to participate in the study after the decision has been made that cochlear implantation or stapes surgery is needed. The treating physician will provide the patient with the information letter and informed consent form. The investigator will obtain informed consent and the treating physician will not be present when signing the informed consent form (S1 File). Patients consent to the use of their data for the research purposes outlined in this protocol, which includes publication of the results once the study has been completed. Patients will not receive financial compensation for participation in the study. Patients can leave the study at any time and for any reason if they wish without any consequences. In case of withdrawal, the patient will remain under the care of their own ENT surgeon, continues with standard medical treatment and will not be followed. Only the adverse events will be monitored.

## Outcomes to be measured

The study design is illustrated in Fig 1 [14]. In the hospital, the taste sensation, food preferences and quality of life will be evaluated preoperatively and at one week, six weeks and six months postoperatively using the TST, EGM, MTPRT and three questionnaires. The questionnaires are the AHSP, QoD and a supplementary questionnaire on taste perception. At home, three months after surgery, the taste sensation is evaluated using the three questionnaires. Evaluation of olfactory function will only take place preoperatively and at one week postoperatively using the Sniffin' Sticks in the hospital. All patients will receive all tests during the regular visits to the ENT department to promote participant retention. Patients will be reminded of the appointments via e-mail or letter.

The patient and outcome assessor are blinded to the presence or absence of CTN injury. The outcome assessors are specialized in ENT and therefore trained in assessing the taste

| | STUDY PERIOD | | | | | |
|---|---|---|---|---|---|---|
| | Enrolment | Surgery | Postoperative | | | Close-out |
| **TIMEPOINT** | Wk. -0-8 | Day 0 | Wk. 1 | Wk. 6-8 | Mo. 3 | Mo. 6 |
| **ENROLMENT** | | | | | | |
| Eligibility screen | ✓ | | | | | |
| Informed consent | ✓ | | | | | |
| Surgery (eCRF) | | ✓ | | | | |
| TST | ✓ | | ✓ | ✓ | ✓ | ✓ |
| EGM | ✓ | | ✓ | ✓ | ✓ | ✓ |
| Sniffin' sticks | ✓ | | ✓ | | | |
| MTPRT | ✓ | | ✓ | ✓ | | ✓ |
| Questionnaires (AHSP, QoD, symptoms taste disturbance) | ✓ | | ✓ | ✓ | ✓ | ✓ |
| **ASSESSMENTS** | | | | | | |
| Baseline variables | ✓ | | | | | |
| Primary outcome: *Taste function (TST)* | ✓ | | ✓ | ✓ | | ✓ |
| Secondary Outcomes | ✓ | | | | | |
| *Taste function (EGM)* | ✓ | | ✓ | ✓ | ✓ | ✓ |
| *Taste function (AHSP)* | ✓ | | ✓ | ✓ | ✓ | ✓ |
| *Postoperative taste disturbance* | ✓ | | ✓ | ✓ | ✓ | ✓ |
| *Olfactory function* | ✓ | | ✓ | ✓ | | |
| *Food preferences* | ✓ | | ✓ | ✓ | | ✓ |
| *Quality of life* | ✓ | | ✓ | ✓ | ✓ | ✓ |

**Fig 1. Schedule of enrolment and assessments adapted from the Standard Protocol Items: Recommendations for Interventional Trials (SPIRIT).**

sensation and olfactory function data. The surgeon will complete an electronic case report form (eCRF) postoperatively, including the predictor variable 'CTN injury'. After a patient has completed all evaluations, the outcome assessor and patient will be unblinded, because the outcome assessor will get access to the fulfilled eCRF. Circumstances that require us to unblind the patient or the outcome assessor during the study are not expected.

**CTN injury.** The CTN injury will be assessed using an eCRF. The eCRF consists of six multiple choice questions about the middle ear surgery and CTN injury, which are completed

by the ENT surgeon after surgery. The outcome 'CTN injury' consists of three categories: no touch, stretching and completely sectioned. For example, if the CTN is partly transected, it will be coded as stretched. The ENT surgeon will also classify the CTN of the operated middle ear into one of the five types of the CTN classification of Uranaka et al. [26]. The classification will be based on the judgement of the ENT surgeon after elevation of the tympanomeatal flap.

**Taste function.** The taste sensation will be assessed using the TST, EGM and AHSP questionnaire.

The validated TST (Burghart Messtechnik, Wedel, Germany) consists of filter paper strips, which are impregnated with 4 concentrations of sweet (0.4, 0.2, 0.1, and 0.05 g/mL sucrose), sour (0.3, 0.165, 0.09, and 0.05 g/mL citric acid), salty (0.25, 0.1, 0.04, and 0.016 g/mL sodium chloride) and bitter (0,006, 0.0024, 0.0009, and 0.0004 g/mL quinine hydrochloride). The strips will be presented in increasing concentrations in a randomized order to each side of the anterior tongue. The patient has to identify which one of the four flavours the taste strip is. The patient has to keep the tongue out of the mouth with the taste strip on the tongue, until he or she has made his decision by pointing to one of the five descriptors (sweet, sour, salty, bitter or tasteless) of the list. The total score of identified taste strips ranges from 0 to 16 and the subscores for each taste ranges from 0 to 4 for each side of the tongue. A higher score indicates a better taste function. In 2003, the TST has been validated in healthy German adults [27].

The Electrogustometer SI-03 (Sensonics Inc., Haddon Heights, NJ) will be used to assess the electric taste. Different studies have established that the EGM is a good clinical tool for measuring taste detection thresholds [28–30]. The electric taste threshold will be detected with anodal stimulation. From 0.25 to 450 uA will be stimulated using the forced-choice single staircase test procedure. The lowest number of uA that generates an electric taste will be determined to be the value of the taste threshold of the CTN [31, 32]. The taste detection threshold will be measured on the lateral region, the innervation region of the CTN, of both sides of the tongue. The lateral regions of the tongue are each two centimetres away from the tip. Stimulus duration affects the threshold values and therefore the stimulus duration will be set at 0.5 sec. for all patients [33, 34]. The AHSP questionnaire consists of 33 questions answered on a 5-point Likert scale and assesses self-judgement of taste and smell perception. In 2001, the ASHP was translated to Dutch and validated in Dutch elderly [35].

**Olfactory function.** The Sniffin' Sticks Screening 12 test (Burghart Messtechnik, Wedel, Germany) will be used to test the olfactory function. This test consists of 12 Sniffin' Sticks, which are pen-like devices filled with liquids containing different odours. After the presentation of the Sniffin' Sticks, the patient has to choose the correct scent description from a list of four scents. The odour identification score ranges from 0 to 12 [36, 37]. The olfactory function will be tested, because the perception of flavour is generated by taste and smell sensation. With this information it will be possible to conclude whether the olfactory function is intact or not [38]. This test is part of regular clinical practice at the UMCU and has been validated in 2008 in healthy Dutch adults [37].

**Food preferences.** The MTPRT will be used to test food preferences. The test consists of pictures of products from four macronutrient categories. These four macronutrient categories are high-carbohydrate, high-fat, high-protein and low-energy. The MTPRT consists of three parts: practicing, liking and ranking. During the practicing part four combinations of four pictures are presented and patients will be asked to rank these pictures according to 'what they most desire to eat at that moment'. The practicing part is intended to familiarize patients with the ranking task. During the liking part 32 pictures of products are presented with the question 'How much do you like [product name]?'. Patients will be asked to rate this question on a 100-point visual analogue scale anchored by 'do not like at all' and 'like extremely'. During the ranking part patients will be asked to rank according to 'what they most desire to eat at that

moment'. Participant will first click on the product they most desire to eat at the moment of completing the task. Afterwards, they will click on the second most desired product, followed by the third most desired product and the product they least desired to eat at the moment of completing the task. In 2017, the MTPRT has been validated in healthy Dutch adults [39].

**Quality of life.**   The translated QoD will be used to assess quality of life. The QoD consists of 24 statements or questions about quality of life ranked by one of the following four options: agree, agree partly, disagree partly, disagree. Two questions about depression and anxiety require a yes or no answer. Nine questions answered on a 10-point Likert scale are used to assess the impact of loss of taste function on health-related quality of life. This questionnaire is based on the questionnaire of olfactory disorders, which has been validated in English patients suffering from olfactory disorders [40–42].

**Symptoms of postoperative taste disturbance.**   A supplementary questionnaire on taste will be used to assess the existence of postoperative symptoms of taste disturbance on a 10-point Likert scale. The symptoms are taste loss, metallic taste sensation, dryness of the mouth, tingling sensations of the tongue, burning sensation of the tongue and numbness of the tongue. The supplementary questionnaire also consists of 11 questions about possible factors that may influence taste sensation.

## Data collection and management

Data handling and protection is conducted according to the ISO standards (27001 & 9001), ICH-GCP and applicable regulations. All data will be handled confidentially and patient information will not be disclosed to third parties. The original signed informed consent forms will be kept in a binder in a *locked* closet in a locked room at the ENT department. The patient will receive a unique identifier, after which members of the research team will extract all necessary clinical parameters into an eCRF of the endorsed online database Castor Electronic Data Capture (Amsterdam, the Netherlands). Castor EDC is a browser-based, metadata-driven EDC software solution and workflow methodology for building and managing online databases. After data collection, the results will be entered into IBM SPSS statistics version 27.0 (IBM Corp., Armonk, NY, USA). Access to databases is password protected and only research members directly involved in this study will have access to all data collected. In order to reproduce the study findings and to help future users to understand and reuse the (meta)data, all changes made to the raw data, including analysis steps will be documented in an analysis plan. All research data will be archived for 15 years after the study has ended.

The data of this research will be used for publication in peer-reviewed international journals. Both positive and negative study results will be disclosed. If the journals do not consider our results for publication, the research will be disclosed through trial registers, websites or databases. Data sharing will be considered upon reasonable request.

## Data analysis plan

IBM SPSS Statistics 27.0 IBM Corp., Armonk, NY, USA) software will be used for the statistical analyses. A test for normality and visual assessment of histograms will be used to assess whether variables are normally distributed. Baseline characteristics will be presented as medians and averages. Mean (or the median if data are not normally distributed) and standard deviation (or the range if data are not normally distributed) will be reported for all continuous outcome parameters. Percentages will be reported when outcome parameters are dichotomous. Parametric methods will be used if data are normally distributed and non-parametric methods will be used if assumption of normality is violated.

**Primary study parameter.** The primary study parameter will be the postoperative identified total taste score of the ipsilateral tongue part of the operated ear, in order to assess the difference in taste sensation between the different groups of CTN injury. This outcome will be presented continuous with a range from 1 to 16 identified taste strips. The effect of CTN injury on the amount of identified taste strips will be evaluated using linear regression analysis. Multicollinearity will be considered by assessing the Variation Inflation Factor (VIF) for each independent variable in SPSS and effect modification will be considered by introducing interaction terms into the multivariable model (predictor*potential effect modifier). Univariable and multivariable analysis will be performed. The following predictor variables will be included in these analysis: age, gender, smoking, weight, gastroesophageal reflux, olfactory dysfunction, salivary hypofunction, medication, rhinosinusitis/allergic rhinitis, previous COVID-19 and diabetes mellitus [15–25].

**Secondary study parameters.** The secondary study parameters are all continuous outcome measures, with the exception of postoperative symptoms of taste disturbance. The latter is a dichotomous outcome. Linear regression analyses will be performed to analyze differences in continuous outcomes between the different categories of CTN injury. Logistic regression analysis will be used to analyze the dichotomous outcome symptoms of postoperative taste disturbance. Multicollinearity and effect modification will also be considered.

**Missing data.** Limited missing data is expected, but the percentage of missing values will be reported and the reasons for missing data will be investigated Potentially missing variables will be handled using multiple imputation.

## Ethics approval and consent to participate

The study will be conducted according to the principles of the Declaration of Helsinki (2013, Fortaleza), 'gedragscode gezondheidsonderzoek' and 'toetsingscriteria eenvormige toetsing' that consist of the laws: 'WMO' (Wet Medisch-wetenschappelijk Onderzoek met mensen), 'WGBO' (Wet op de Geneeskundige Behandelings Overeenkomst), 'AVG' (Algemene Verordening Gegevensbescherming) and 'EU GDPR' (General Data Protection Regulation). The study protocol was approved by the Institutional Review Board of the UMCU (protocol 21–466 V.2; September 2021). All substantial amendments will be notified to the Institutional Review Board of the University Medical Center Utrecht. Informed consent will be obtained from all participating patients before the first measurements.

## Trial steering committee

The quality of the study, including speed of inclusion, drop-out rate and the presence and completeness of the monocentric study, will be independently monitored once a year by a local monitor (UMCU). The local monitor will check all data of the first five participants, 10% of the signed informed consent forms and the study file. The Institutional Review Board classified this study as low risk because all interventions are standard medical care. Therefore, no data safety monitoring committee (DSMC) is needed and no interim analyses will be performed during this trial. Once a year, a summary of the trial progress will be submitted to the local Institutional Review Board. Information will be provided: the date of inclusion of the first subject, the number of participants enrolled, the number of participants who completed the trial, serious adverse events, other problems and amendments.

## Discussion

To date, there is no consensus regarding which type of CTN injury (sacrificing or stretching) leads to the least burden for the patient. The level of evidence from previously published

studies on this subject is low, as these studies have a high risk of bias. This study is the first to validate and quantify the effect of chorda tympani nerve injury on taste function. The findings of this study may lead to evidence-based proof of the effect of chorda tympani injury on taste function with consequences for surgical strategies. Another strength of this study is that a quality of life questionnaire (QoD) will be combined with taste measurements (EGM and TST).

A limitation of our study is that not all outcome measures will be tested during the three-month follow-up measurement after surgery, to ensure patient compliance. Secondly, the QoD and the supplementary questionnaire on postoperative taste disturbance have not been validated in a Dutch population and the QoD was designed to primarily assess olfactory dysfunction and not taste dysfunction. However, the QoD does contain questions concerning the influence of taste disturbance on quality of life and there is no (validated) alternative tool for assessing taste-related quality of life.

## Supporting information

**S1 File. Informed consent form TACO study.**
(DOCX)

**S2 File. Protocol TACO study V.2 2021.**
(DOCX)

**S3 File. Completed SPIRIT checklist TACO study.**
(PDF)

## Author Contributions

**Conceptualization:** Esther E. Blijleven, Inge Wegner, Robert J. Stokroos, Hans G. X. M. Thomeer.

**Writing – original draft:** Esther E. Blijleven, Inge Wegner, Robert J. Stokroos, Hans G. X. M. Thomeer.

**Writing – review & editing:** Esther E. Blijleven, Inge Wegner, Robert J. Stokroos, Hans G. X. M. Thomeer.

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
