## [Decision Letter · Decision Letter 0]

20 Dec 2022

PONE-D-22-22179The impact of injury of the chorda tympani nerve during primary stapes surgery or cochlear implantation on taste function, quality of life and food preferences: a study protocol for a prospective prognostic trialPLOS ONE

Dear Dr. Blijleven,

Thank you for submitting your manuscript to PLOS ONE. After careful consideration, we feel that it has merit but does not fully meet PLOS ONE’s publication criteria as it currently stands. Therefore, we invite you to submit a revised version of the manuscript that addresses the points raised during the review process. Your manuscript has been assessed by two peer-reviewers and their reports are appended below.  The reviewers' comment that your study would be strengthened by additional details and clarifications, for example for the study design, guidelines, and statistical analysis. In addition, the reviewers have made suggestions on the data collection and exclusion criteria proposed in this study protocol.  The reviewers have recommended a number of citations as a part of their review. We would recommend that you thoroughly evaluated these requested references and determine whether the articles are relevant to the current study. You may feel free to disregard references with tangible relevance to the study reported in the manuscript. Could you please revise the manuscript to carefully address the concerns raised?

We look forward to receiving your revised manuscript.

Kind regards,

Maria Elisabeth Johanna Zalm, Ph.D

Editorial Office

PLOS ONE

Journal Requirements:

Reviewers' comments:

Reviewer's Responses to Questions

**Comments to the Author**

1. Does the manuscript provide a valid rationale for the proposed study, with clearly identified and justified research questions?

Reviewer #1: Partly

Reviewer #2: Yes

2. Is the protocol technically sound and planned in a manner that will lead to a meaningful outcome and allow testing the stated hypotheses?

Reviewer #1: Yes

Reviewer #2: Yes

3. Is the methodology feasible and described in sufficient detail to allow the work to be replicable?

Reviewer #1: Yes

Reviewer #2: Yes

4. Have the authors described where all data underlying the findings will be made available when the study is complete?

Reviewer #1: No

Reviewer #2: Yes

5. Is the manuscript presented in an intelligible fashion and written in standard English?

Reviewer #1: No

Reviewer #2: Yes

6. Review Comments to the Author

You may also provide optional suggestions and comments to authors that they might find helpful in planning their study.

Reviewer #1: This is an interesting double-blind prognostic study, designed to investigate the effect of CTN injury on postoperative taste function, defined as the postoperative taste strip score, in patients undergoing primary stapes surgery or primary cochlear implantation.

1. This has been stated to be a prognostic study- can the authors be more explicit in terms of design , i.e whether this is a description, association, prediction and causation, from the stats methods, its suggestive of either description or association.

2. And consider whether these guidelines need to be used; PROGRESS, TRIPOD and CHARMS

3. Pre-specified covariates have been stated to include 14 variables. In the statistical analysis section, it would make sense to include univariate analysis of these covariates.

4. In addition state, are the authors going to consider multicollinearity?

5. Any consideration of moderators or mediating factors?

6. In the statistical analysis section, it might also be useful to include that, if assumption of normality are violated for the continuous outcome i.e the primary outcome - non parametric methods will be used.

Reviewer #2: The reviewer thanks for the opportunity to analyze this work. It is a good written protocol. However, I have some suggestions:

- Converted all “we…” in passive form.

- Ear malformation should be considered in the exclusion criteria

- Better describe surgical procedure performed and the type of method (endoscopic or microscopic).

Read: (doi:10.1177/0194599821990669; DOI: 10.5152/iao.2017.3322; DOI: 10.7874/jao.2021.00388).

- Eventual other post-surgical complications should be collected.

- Why will you not perform some valuations at all follow-up times? for example, the olfactory function evaluation?

7. PLOS authors have the option to publish the peer review history of their article (what does this mean?). If published, this will include your full peer review and any attached files.

Reviewer #1: No

Reviewer #2: No

---

## [Author Response · Author response to Decision Letter 0]

8 Feb 2023

Dear Dr. Maria Elisabeth Johanna Zalm,

Thank you for considering our manuscript. Enclosed you will find our answers to the questions. 

Yours sincerely, 

Esther E. Blijleven, Robert J. Stokroos, Inge Wegner and Hans G.X.M. Thomeer

Journal Requirements

Answer: Thank you for your comments. We have changed the filenames and the manuscript conforms to the style requirements. 

b) If there are no restrictions, please upload the minimal anonymized data set necessary to replicate your study findings as either Supporting Information files or to a stable, public repository and provide us with the relevant URLs, DOIs, or accession numbers. For a list of acceptable repositories, please see http://journals.plos.org/plosone/s/data-availability#loc-recommended-repositories. We will update your Data Availability statement on your behalf to reflect the information you provide. 

Answer: This manuscript is a study protocol. We cannot share any data because we do not have results yet. The intention is to submit the manuscript containing the results of this study to PLOS one. We will publish an anonymized data set of the results of our research at that time.

Answer: We have moved the ethics statement from the declaration section to the methods section. (see page 15-16, line 379-388)

Answer: We have included a caption of the supporting information files at the end of our manuscript. (see page 20, line 496-499)

Reviewer 1

This is an interesting double-blind prognostic study, designed to investigate the effect of CTN injury on postoperative taste function, defined as the postoperative taste strip score, in patients undergoing primary stapes surgery or primary cochlear implantation. 

Answer: Thank you for your comments.

1. This has been stated to be a prognostic study- can the authors be more explicit in terms of design , i.e whether this is a description, association, prediction and causation, from the stats methods, its suggestive of either description or association.

Answer: The study is a prognostic association study. Addition manuscript: Double-blind prospective prognostic association study (see page 1, line 4-5; page 3, line 54 and page 7, line 146)

2. And consider whether these guidelines need to be used; PROGRESS, TRIPOD and CHARMS 

Answer: We will not make a prediction model with the results, so the PROGRESS, TRIPOD, CHARMS guidelines are not applicable to our study. 

3. Pre-specified covariates have been stated to include 14 variables. In the statistical analysis section, it would make sense to include univariate analysis of these covariates. 

Answer: We will also perform univariate analysis of these covariates. Addition manuscript: Univariable and multivariable analysis will be performed. (see page 15, line 360-361) 

4. In addition state, are the authors going to consider multicollinearity? 

Answer: We will consider (multi)collinearity by assessing the Variation Inflation Factor (VIF) for each independent variable in SPSS. Additon manuscript: Multicollinearity will be considered by assessing the Variation Inflation Factor (VIF) for each independent variable in SPSS. (see page 15, line 357-359)

5. Any consideration of moderators or mediating factors? 

Answer: We will consider effect modification by introducing interaction terms into the multivariable model (predictor*potential effect modifier). Addition manuscript: effect modification will be considered by introducing interaction terms into the multivariable model (predictor*potential effect modifier). (see page 15, line 359-360)

6. In the statistical analysis section, it might also be useful to include that, if assumption of normality are violated for the continuous outcome i.e the primary outcome - non parametric methods will be used. 

Answer: We will use non-parametric methods if assumption of normality is violated. Addition manuscript: Parametric methods will be used if data are normally distributed and non-parametric methods will be used if assumption of normality is violated. (see page 14, line 349-350)

Reviewer 2

The reviewer thanks for the opportunity to analyze this work. It is a good written protocol. 

Answer: Thank you for your comments. 

1. Converted all “we…” in passive form. 

Answer: We have converted all text from active to passive. (see manuscript) 

2. Ear malformation should be considered in the exclusion criteria.

Answer: Ear malformation is an exclusion criteria. Addition manuscript: Exclusion criteria: ear malformation. (see page 8, line 175)

3. Better describe surgical procedure performed and the type of method (endoscopic or microscopic).

Read: (doi:10.1177/0194599821990669; DOI: 10.5152/iao.2017.3322; DOI: 10.7874/jao.2021.00388). 

Answer: The surgical method will be microscopic for all surgeries. 

4. Eventual other post-surgical complications should be collected. 

Answer: We will collect all adverse events, including all post-surgical complications. 

5. Why will you not perform some valuations at all follow-up times? for example, the olfactory function evaluation? 

Answer: We will not perform the Sniffin’ Sticks test at all follow-up times to limit the burden on patients. The Institutional Review Board of the UMCU closely monitors the burden of patients participating in research and therefore we must be critical of the tests to which we expose the patient. After consultation of the Institutional Review Board of the UMCU we have decided to only perform the Sniffin’ Sticks test twice, because we expect the sense of smell does not change (much) over time.

---

## [Decision Letter · Decision Letter 1]

4 Apr 2023

The impact of injury of the chorda tympani nerve during primary stapes surgery or cochlear implantation on taste function, quality of life and food preferences: a study protocol for a double-blind prospective prognostic association study

PONE-D-22-22179R1

Dear Dr. Blijleven

We’re pleased to inform you that your manuscript has been judged scientifically suitable for publication and will be formally accepted for publication once it meets all outstanding technical requirements.

Kind regards,

Annalisa Pace

Academic Editor

PLOS ONE

Reviewers' comments:

Reviewer's Responses to Questions

**Comments to the Author**

1. Does the manuscript provide a valid rationale for the proposed study, with clearly identified and justified research questions?

Reviewer #1: Yes

Reviewer #3: Partly

2. Is the protocol technically sound and planned in a manner that will lead to a meaningful outcome and allow testing the stated hypotheses?

Reviewer #1: Yes

Reviewer #3: Partly

3. Is the methodology feasible and described in sufficient detail to allow the work to be replicable?

Reviewer #1: Yes

Reviewer #3: Yes

4. Have the authors described where all data underlying the findings will be made available when the study is complete?

Reviewer #1: No

Reviewer #3: Yes

5. Is the manuscript presented in an intelligible fashion and written in standard English?

Reviewer #1: Yes

Reviewer #3: Yes

6. Review Comments to the Author

You may also provide optional suggestions and comments to authors that they might find helpful in planning their study.

Reviewer #1: I have no further comments.

All comments have been addressed.

Reviewer #3: Dear authors,

this is NOT the first study to validate quantify the efect of CT nerve injury on taste function. Chek and include papers of Perez R, Božanić Urbančič N.

The electrogustometry is not a standard testing procedure; how are you going to perform it "on one side of the tounge"?

7. PLOS authors have the option to publish the peer review history of their article (what does this mean?). If published, this will include your full peer review and any attached files.

Reviewer #1: No

Reviewer #3: **Yes: **Saba Battelino

---

## [Editor Report · Acceptance letter]

9 May 2023

PONE-D-22-22179R1 

The impact of injury of the chorda tympani nerve during primary stapes surgery or cochlear implantation on taste function, quality of life and food preferences: a study protocol for a double-blind prospective prognostic association study 

Dear Dr. Blijleven:

I'm pleased to inform you that your manuscript has been deemed suitable for publication in PLOS ONE. Congratulations! Your manuscript is now with our production department. 

Kind regards, 

on behalf of

Dr. Annalisa Pace 

Academic Editor

PLOS ONE